# Biomonitorization of concentrations of 28 elements in serum and urine among workers exposed to indium compounds

Nan Liu[1], Yi Guan[1], Bin Li[2], Sanqiao Yao [1,3]*

1 School of Public Health, North China University of Science and Technology, Tangshan, Hebei, China,
2 Institute for Occupational Health and Poison Control in China Center for Disease Prevention and Control,
Beijing, China, 3 Xinxiang Medical University, Xinxiang, China

* sanqiaoyao@xxmu.edu.cn

Biomonitorization of concentrations of 28 elements
in serum and urine among workers exposed to
indium compounds. PLoS ONE 16(2): e0246943.

CHINA

**Data Availability Statement:** All relevant data are
within the manuscript and its Supporting
Information files.

**Funding:** This study was supported by National
Public Welfare Health Industry Scientific Research

## Abstract

Many studies have documented the abnormal concentrations of metals/metalloids in serum
or urine of occupational workers, but no works systematically analysed the concentrations
of elements in serum or urine of indium-exposed workers. This study was aimed to assess
28 elements in serum and urine from 57 individuals with occupational exposure to indium
and its compounds. Control subjects were 63 workers without metal exposure. We collected
information on occupation and lifestyle habits by questionnaire. Biological samples were col-
lected to quantify elements by inductive coupled plasma-mass spectrometer. Air in the
breathing zones was drawn at flow rates of 1.5–3 L/min for a sampling period of 6 to 8 h,
using a Model BFC-35 pump. The average ambient indium level was 0.078 mg/m$^3$. Serum/
urine Indium levels were significantly higher in indium-exposed workers than in controls ($P <$
0.01). Moreover, serum/urine indium concentrations in the group with 6–14 years and $\geq$15
years of employment were significantly higher than those with $\leq$5 employment years($P <$
0.05). Ten of the other 27 elements/metals measured were higher in serum/urine in indium-
exposed workers compared to the controls (aluminum, beryllium, cadmium, cesium, chro-
mium, lithium, manganese, magnesium, molybdenum and vanadium). Zinc levels in serum/
urine were significantly decreased in the indium-exposed workers. Additionally, other ele-
ments/metals were higher in one specimen (serum or urine) but lower in the other (Selenium
was lower in serum but higher in urine in the indium-exposed workers compared with the
controls; likewise Thallium and Rubidium were higher in serum but lower in urine). Linear
regression analyses, revealed significant correlations between serum and urine for indium,
aluminum, arsenic, barium, cadmium, cesium, cobalt, selenium, silver, and zinc ($P <$ 0.05).
These data suggest that occupational exposure to indium and its compounds may disturb
the homeostasis of trace elements in systemic circulation, indium concentrations in serum
or urine appear reflective of workers' exposure to ambient indium and their years of working,
respectively. The serum/urine levels of essential metals are modified by exposure to indium
in occupationally exposed workers. Further studies including larger sample size and more
kinds of biological sample are needed to validate our findings.

(No. 201402021) and Graduate Innovation Project (No. 2019B17).

**Competing interests:** All authors declare that there are no conflicts of interest regarding the publication of this article.

## Introduction

Indium is relatively rare element found in ores of zinc, copper, and tin and has been used in flat-panel displays, optoelectronic devices, and photovoltaic cells for decades. Occupational exposure to indium and its compounds can result in potentially fatal indium lung disease including pulmonary alveolar proteinosis that may progress to fibrosis [1–7]. In the process of indium smelting, there are occupational hazards such as dust, lead, arsenic, cadmium, indium, zinc, hydrogen arsenide, various acids and bases, noise, high temperature. Workers are likely to be exposed to various metals/metalloids and elements simultaneously in their working environment. Lead, cadmium, and arsenic were reported to be the main hazardous metals/metalloids. They cause serious damage to many target organs in human bodies through the mechanism of inflammation, production of oxidative stress, and interference with essential elements. The interactions among these toxic metals/metalloids are extremely complex and some effects which have not been observed in single constituent exposure may occur. On the other hand, copper, and zinc are essential trace elements with regulatory, immunologic, and antioxidant functions, but when they are beyond or below a certain serum concentration, they may cause a threat to human health. Under the exposure to multiple metals/metalloids which may compete with or regulate the function of the essential elements in human bodies, they may cause additive, synergistic, or antagonistic effects, or even new effects may occur [8]. To date, no studies on indium occupational workers concerned trace elements. Indium pulmonary toxicity may be associated with its interaction with other essential trace elements. Although there are some reports on levels of indium in blood or urine during production process, the question as to how low-level, long-term exposure to indium and its compounds may affect blood or urine levels of trace metals is unanswered [9]. Therefore, it is important to know and measure trace elements status in indium-exposed workers because the alterations in the content may play an important role in the pathogenesis of pulmonary alveolar proteinosis and related metabolic risk factors.

This study will analyze 28 elements in two different human biological materials (serum/urine) by inductively coupled plasma-mass spectrometer (ICP-MS), and assess the classical fluids (serum and urine) for biomonitoring the internal dose of individuals occupationally exposed to indium and its compounds. Because the indium-exposed workers were typically population occupationally exposed to indium compounds and toxic metals, they are particularly suitable for biological monitoring programs to measure trends of occupational exposure with early biological effects and to explore their dose-response relationships. In particular, a risk assessment methods should be also adopted, given that this population may present an increased health risk to pulmonary alveolar proteinosis disease. Hence this study provides values that may be useful for comparisons in future studies, or in addressing occupational health challenges associated with indium and its compounds exposure. The purpose of this study was to investigate whether or not chronically occupational exposure to indium and its compounds altered the homeostasis of essential trace elements in biological fluids.

## Subjects and methods

### Study population

**Subjects.** This was an exploratory study that used data obtained during a health check by a cross-sectional study between February 21 and March 15, 2015. The subjects were workers in indium ingot production plant from Guangxi, China who were mainly exposed to indium metal with a potentially high level, but also to $In_2SO_4$, $In_2O_3$, and to a lesser extent, indium chloride [$InCl_3$] with a potentially high level of exposure. During operation, the indium-

exposed workers wore dust-free clothes, while maintenance workers wore full face filter cotton, goggles and protective gloves. The control subjects in this study were a random sample of all candidates derived from another nearby factory, who were office workers and had no history of occupational exposure to indium and other metals. The selected control subjects were matched to the age, gender, average employment history, smoking status and drinking habits with indium-exposed workers.

The selection criteria of the study population for this study were:(1) no history of occupational exposure to other metals; (2) no supplement of trace elements; (3) complete information of outcome measured value (both in serum and urine); and (4) excluding outliers of each measured value. In this study, new occurrence of hypomagnesemia was defined as outliers, and thus observation in worker who already had outliers was excluded. Finally, the data analysis included a total of 120 subjects, which comprised of 57 indium-exposed workers and 63 controls.

**Patient and public involvement.** Characteristics of the sampling method used to collect the biological samples selected for this study have been described previously [10]. An interview with a questionnaire was completed by trained interviewers to obtain detailed information on personal information, occupational history, job description, lifestyle information (smoking, alcohol consumption, dietary habits), and personal medical history. Written consent was obtained from the subjects who participated in this study and they were also informed of their right to withdraw anytime. The proposal was previously approved by the Ethics Review Committee of the North China University of Science and Technology (Approval No.15080). All study participants in both indium occupational workers and control groups at the time of interview had no reported exposure to other toxic substances, radiation therapy, autoimmune disease, or substance abuse (such as antibiotics).

**Potential confounders and sources of bias.** As potential confounding factors, data on age, gender, smoking status, and drinking habits were obtained from the interview with a questionnaire. We collected data of indium-handling workers who wore protective equipment, but controls have no data for protective equipment. In addition, the missing and incorrect values (i.e., new occurrence of hypomagnesemia, among others) were reconfirmed with indium ingot production plant and another nearby factory, and tried to exclude selection bias.

## Sample preparation and analysis

**Workplace air concentrations of indium dust.** Based on the indium production process, four different locations in the workplace were identified as the monitor sites. Air samplers were placed in each workplace. The workplace air was drawn at a flow rate of 1.5-3L/min for a sampling period of 6 h or 8 h, except during lunchtime, using a Model BFC-35 pump equipped with a mixed cellulose ester membrane filter (a diameter of 37 mm, a cut size of 0.8 μm). The samples were collected for two more times at each monitoring site. Workers who wore the personal sampling devices were asked to record the contents of their jobs during the work periods.

A glass fiber filter on which indium-containing particulate matter had been collected was placed into a flask containing 20 ml of mixed acid (nitric acid: ultrapure water = 1:1) and subjected to ultrasonic application for 60 min, and digested on a hot plate at 120˚C for 60 min. The digested sample was diluted to 10ml with ultrapure water and then subjected to analysis. All analyses were performed in a blinded manner, and the results were confirmed by replication on the following day.

A standard solution of indium for inductively coupled plasma mass spectrometer (ICP-MS) (Agilent 7500a, Agilent Technology, USA) was used for preparation of a calibration curve for

the quantitation. The quantitation limit of the airborne indium concentration was estimated to be 0.045 μg In/m$^3$. This estimate is ~15% of the acceptable indium exposure limit of 0.0003 mg/m$^3$.

**Sample preparation and elements analysis.** Sample collection and processing were carried out in local clinics. Five milliliters of venous blood was collected in a pro-coagulation tube, let stand at room temperature for 15 min, and centrifuged at 12,000 rpm for 10 min to separate serum, and immediately transferred to 2 mL frozen pipe. A spot 10-mL urine sample collected from each subject at the end of a work shift have been described previously [10]. All samples were stored at -80˚C until analysis. All test tubes used in the study were free of metal contamination.

The 28 elements analyzed in this work were aluminum (Al), arsenic (As), barium (Ba), beryllium (Be), bismuth (Bi), cadmium (Cd), calcium (Ca), cesium (Cs), chromium (Cr), cobalt (Co), copper (Cu), indium (In), iron (Fe), lead (Pb), lithium (Li), manganese (Mn), magnesium (Mg), molybdenum (Mo), nickel (Ni), potassium (K), rubidium (Rb), selenium (Se), silver (Ag), sodium (Na), strontium (Sr), thallium (Tl), vanadium (V), and zinc (Zn).

At the time of sample analysis, serum and urine samples were brought to room temperature. An aliquot of 500 μL serum and urine samples was diluted with a solution containing 0.1% (V/V) Triton-X-100 (Sigma, USA) and 1% ultrapure concentrated nitric acid (Sigma, USA) to a 5 mL total volume. Prior dilution of each sample was critical in order to obtain the best results. The samples were then quantified by ICP-MS using freshly made multi-element stock solution on the day of analysis. The instrument parameters are as followed: nebulizer carrier gas flow, 1.10 L/min; sample depth, 4.8 mm; RF power, 1450 W; sampler/skimmer, nickel; CeO$^+$/Ce$^+$, <0.5%; $^{140}$Ce$^{16}$O/$^{140}$Ce, <2%. Calibration was performed using a certified reference standard (Agilent, USA). Validity of the calibration curve was evaluated by analyzing the standards from the same source after every two hours injection. The calibration curve was considered valid if the observed concentration of the independent standard was within 10% of the expected concentration. We also measured the recovery of each sample at random. The internal standard and the limits of detection (LOD) for these 28 elements were shown in Table 1.

## Statistical analyses

All statistical analyses were conducted with SPSS 17.0 software. All data are presented as the mean ± SD unless otherwise stated. Differences between two groups were evaluated

**Table 1. The limits of detection for these 28 elements.**

| Element | Internal standard | Limits of detection (LOD) | Element | Internal standard | Limits of detection (LOD) |
|---|---|---|---|---|---|
| $^{27}$Al | $^{45}$Sc | 0.108 μg/L | $^{7}$Li | $^{45}$Sc | 0.025 μg/L |
| $^{85}$As | $^{72}$Ge | 0.032 μg/L | $^{55}$Mn | $^{45}$Sc | 0.014 μg/L |
| $^{137}$Ba | $^{103}$Rh | 0.041 μg/L | $^{24}$Mg | $^{45}$Sc | 0.795 μg/L |
| $^{9}$Be | $^{45}$Sc | 0.007 μg/L | $^{95}$Mo | $^{89}$Y | 0.057 μg/L |
| $^{209}$Bi | $^{103}$Rh | 0.014 μg/L | $^{60}$Ni | $^{72}$Ge | 0.042 μg/L |
| $^{114}$Cd | $^{103}$Rh | 0.075 μg/L | $^{39}$K | $^{45}$Sc | 4.220 μg/L |
| $^{40}$Ca | $^{45}$Sc | 1.202 μg/L | $^{89}$Rb | $^{89}$Y | 0.053 μg/L |
| $^{133}$Cs | $^{103}$Rh | 0.061 μg/L | $^{82}$Se | $^{89}$Y | 0.142 μg/L |
| $^{52}$Cr | $^{45}$Sc | 0.039 μg/L | $^{108}$Ag | $^{103}$Rh | 0.036 μg/L |
| $^{59}$Co | $^{45}$Sc | 0.015 μg/L | $^{23}$Na | $^{45}$Sc | 2.560 μg/L |
| $^{63}$Cu | $^{72}$Ge | 2.526 μg/L | $^{88}$Sr | $^{89}$Y | 0.009 μg/L |
| $^{115}$In | $^{103}$Rh | 0.053 μg/L | $^{205}$Tl | $^{103}$Rh | 0.037 μg/L |
| $^{56}$Fe | $^{45}$Sc | 4.068 μg/L | $^{51}$V | $^{45}$Sc | 0.079 μg/L |
| $^{208}$Pb | $^{103}$Rh | 0.019 μg/L | $^{66}$Zn | $^{72}$Ge | 2.431 μg/L |

for significance using Welch's *t* test for datas. The differences between two means were analyzed by independent samples T test. Associations between serum and urine concentrations of elements were analyzed by a linear regression. The standardized regression coefficient was used to compare the effect of different independent variables (serum and urine for indium, aluminum, arsenic, barium, cadmium, cesium, cobalt, selenium, silver, and zinc) on dependent variables. To relate indium concentration with trace element concentrations variables, we used multiple linear regression and included age, length of service, gender, smoking and alcohol consumption as co-variates. Age and length of service were adjusted as continuous random variables, while gender, smoking and alcohol consumption were categorical variables. The results which includes all covariates with p-values less than 0.05 are reported. All statistical tests are two-sided with a significance level of 0.05.

## Results

### Demographics of the study population

The data of 138 subjects (included 69 indium-exposed workers and 69 control subjects) were analyzed. The subjects selection flowchart are summarized in Fig 1. Initially, 8 indium-exposed workers and 5 controls were excluded owing to oral calcium tablets and oral iron supplement; 4 workers with missing biological samples were then excluded. One observation had outliers outcome were excluded. Finally, data of elemental analysis from 120 workers of serum and urine were analyzed.

The demographic characteristics are summarized in Table 2. The distribution of smoking status was significantly different between the 57 indium-exposed workers and 63 unexposed controls. The exposed workers included less smokers than the controls. There were no significant differences in age, sex distribution, average employment history and drinking habits between the exposed workers and controls.

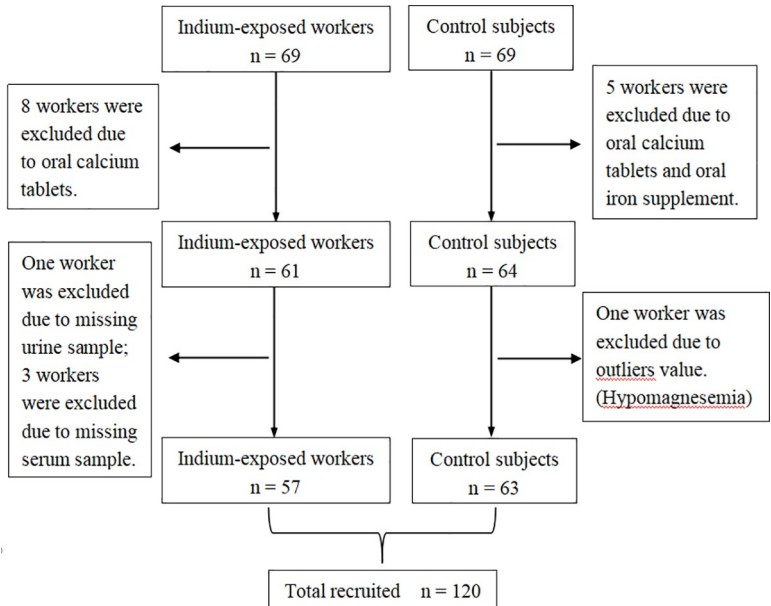

**Fig 1. Flow of indium-exposed workers and unexposed workers in the study.**

**Table 2. Demographics of the study population.**

| | | Controls (n = 63) | Workers (n = 57) |
|---|---|---|---|
| Age (years) | Mean ± SD | 39.32 ± 11.37 | 37.82 ± 9.34 |
| Sex | Male | 46 (73.0) | 41 (71.9) |
| | Female | 17 (27.0) | 16 (28.1) |
| Smoking | Non-smoker | 32 (50.8) | 36 (63.2) |
| | Smoker | 31 (49.2) | 21 (36.8) |
| Drinking | Non-drinker | 43 (68.3) | 39 (68.4) |
| | Drinker | 20 (31.7) | 18 (31.6) |
| Employment period (years) | Mean ± SD | 8.97 ± 8.39 | 8.31 ± 7.54 |

## Ambient indium concentrations

The airborne indium levels in the indium ingot production plant varied from 0.001 to 1.12 mg/m$^3$ (median indium level, 0.008 mg/m$^3$). The average ambient air level of indium was 0.078 ± 0.065 mg/m$^3$. In the control group, however, none of the airborne samples reached a quantifiable level of indium.

## Determination of 28 elements in serum and urine

The distribution of indium and other 27 elements levels in serum is shown in Table 3. Serum concentrations of the 28 elements varied from 26.38 ng/L (Be) to 346.72 mg/L (Na). Eleven of the 28 (39%) elements showed approximately equal concentrations in serum of the workers and controls. These included As, Ba, Bi, Ca, Co, Fe, Ni, K, Ag, Na and Sr. The serum concentrations of indium in workers and controls were (39.26±25.57) and (4.93±2.76) µg/L ($P$ <0.01), respectively. Analyses revealed that there were 28% increase in serum concentration of Al, 31% increase in Be, 27% increase in Cd, 24% increase in Cs, 55% increase in Cr, 19% increase in Cu, 50% increase in Pb, 16% increase in Li, 35% increase in Mn, 7% increase in Mg, 15% increase in Mo, 23% increase in Rb, 36% increase in Tl, 27% increase in V, and 29% decrease in Zn, 19% decrease in Se among indium-exposed workers in comparison to those in controls.

**Table 3. Concentrations of 28 elements in serum and urine of workers and control subjects.**

| Elements | Biological sample | Controls (n = 63) | | | | Workers (n = 57) | | | |
|---|---|---|---|---|---|---|---|---|---|
| | | Mean± SD | Range | Median | 5-95th percentiles | Mean± SD | Range | Median | 5-95th percentiles |
| Al (µg/L) | Serum | 7.82±2.51 | 3.10–13.96 | 7.59 | 4.42–12.94 | 10.01±3.35** | 5.88–21.39 | 9.04 | 6.24–18.41 |
| | Urine | 17.04±5.03 | 1.75–22.56 | 18.77 | 5.10–21.75 | 19.92±6.53** | 4.81–40.16 | 17.91 | 13.19–35.27 |
| As (µg/L) | Serum | 4.73±3.46 | 0.31–14.98 | 4.31 | 0.52–11.12 | 5.40±3.70 | 0.43–13.06 | 5.53 | 0.89–12.42 |
| | Urine | 3.62±3.21 | 0.18–17.86 | 2.85 | 0.34 = 9.86 | 8.33±4.51** | 0.52–19.06 | 8.03 | 1.38–15.90 |
| Ba (µg/L) | Serum | 7.99±6.57 | 2.20–36.59 | 6.12 | 2.59–25.21 | 9.98±5.89 | 3.67–42.22 | 8.97 | 4.27–20.34 |
| | Urine | 3.01±0.87 | 2.10–4.79 | 2.69 | 2.11–4.40 | 3.00±0.83 | 2.14–4.92 | 2.60 | 2.17–4.65 |
| Be (ng/L) | Serum | 26.38±11.84 | 3.80–58.40 | 26.25 | 8.24–48.58 | 34.62±21.59* | 7.10–147.5 | 31.23 | 11.38–69.58 |
| | Urine | 17.75±3.46 | 13.30–29.90 | 17.10 | 13.44–24.54 | 19.80±3.74** | 14.20–29.20 | 19.10 | 14.38–26.50 |
| Bi (µg/L) | Serum | 15.61±13.56 | 1.23–76.93 | 12.71 | 3.12–47.21 | 30.00±26.11 | 3.39–100.43 | 20.78 | 5.03–94.58 |
| | Urine | 130.05±33.51 | 76.80–256.50 | 123.30 | 85.68–198.24 | 150.01±53.77* | 67.50–281.70 | 141.00 | 81.39–249.28 |
| Cd (µg/L) | Serum | 0.77±0.35 | 0.33–1.99 | 0.71 | 0.35–1.46 | 0.98±0.37** | 0.42–1.92 | 0.91 | 0.43–1.65 |
| | Urine | 0.28±0.21 | 0.07–1.13 | 0.20 | 0.08–0.77 | 0.38±0.30* | 0.12–1.67 | 0.25 | 0.14–1.15 |
| Ca (mg/L) | Serum | 104.33±36.23 | 62.15–222.05 | 88.80 | 64.59–206.13 | 94.98±36.85 | 46.05–220.55 | 86.23 | 47.64–187.53 |

*(Continued)*

**Table 3.** (Continued)

| Elements | Biological sample | Controls (n = 63) | | | | Workers (n = 57) | | | |
|---|---|---|---|---|---|---|---|---|---|
| | | Mean± SD | Range | Median | 5-95th percentiles | Mean± SD | Range | Median | 5-95th percentiles |
| Cs (µg/L) | Urine | 80.62±21.84 | 52.19–141.12 | 74.96 | 52.89–124.20 | 94.76±31.62** | 51.34–152.15 | 82.73 | 53.09–150.28 |
| | Serum | 4.33±1.12 | 2.56–8.55 | 4.12 | 2.86–6.61 | 5.35±2.61** | 2.25–16.05 | 4.70 | 2.71–10.78 |
| Cr (µg/L) | Urine | 3.23±1.03 | 2.10–7.24 | 3.23 | 2.14–4.86 | 4.01±1.73** | 2.01–8.84 | 3.50 | 2.09–7.43 |
| | Serum | 0.86±0.31 | 0.43–1.72 | 0.75 | 0.50–1.53 | 1.33±0.53** | 0.43–2.77 | 1.42 | 0.49–2.48 |
| Co (µg/L) | Urine | 1.13±0.87 | 0.20–4.37 | 1.02 | 1.24–3.66 | 2.58±1.38** | 0.31–6.47 | 2.36 | 0.60–5.49 |
| | Serum | 1.54±0.62 | 0.75–3.57 | 1.39 | 0.84–2.48 | 1.75±0.56 | 0.92–3.36 | 1.80 | 0.92–2.63 |
| Cu (µg/L) | Urine | 0.29±0.18 | 0.10–1.05 | 0.26 | 0.10–0.84 | 1.22±0.54** | 0.33–2.60 | 1.19 | 0.35–2.29 |
| | Serum | 0.83±0.14 | 0.65–1.46 | 0.81 | 0.67–1.10 | 0.99±0.32** | 0.68–2.13 | 0.86 | 0.72–1.97 |
| In (ng/L) | Urine | 31.92±12.39 | 10.71–60.26 | 25.87 | 15.13–54.11 | 52.28±25.19** | 10.12–105.70 | 51.13 | 12.43–104.13 |
| | Serum | 4.93±2.76 | 0.93–13.64 | 4.36 | 0.98–10.38 | 39.26±25.57** | 11.86–137.63 | 32.93 | 14.78–97.99 |
| Fe (mg/L) | Urine | 2.34±1.77 | 0.01–8.61 | 1.81 | 0.11–5.02 | 9.24±10.31** | 0.07–54.93 | 7.24 | 0.29–31.90 |
| | Serum | 4.34±0.56 | 2.74–5.88 | 4.37 | 3.11–5.11 | 5.99±1.71 | 3.00–9.68 | 6.05 | 3.57–9.58 |
| Pb (µg/L) | Urine | 1.54±0.55 | 1.03–3.11 | 1.19 | 1.10–2.38 | 2.70±1.88** | 1.11–8.98 | 2.09 | 1.13–8.19 |
| | Serum | 6.86±3.58 | 2.36–21.92 | 6.13 | 2.97–15.43 | 10.29±6.23** | 3.32–38.55 | 8.54 | 3.96–25.12 |
| Li (µg/L) | Urine | 0.97±0.44 | 0.09–2.37 | 1.14 | 0.21–1.37 | 1.07±0.99 | 0.06–7.19 | 1.03 | 0.13–2.33 |
| | Serum | 38.27±14.76 | 29.93–130.00 | 35.10 | 31.07–47.66 | 44.45±7.26** | 34.58–69.80 | 42.30 | 35.39–57.02 |
| Mn (µg/L) | Urine | 37.04±13.72 | 17.05–79.60 | 36.10 | 18.35–68.80 | 49.84±18.96** | 22.35–120.25 | 43.85 | 28.96–85.64 |
| | Serum | 7.41±2.74 | 3.41–15.92 | 7.22 | 3.67–13.76 | 9.97±3.79** | 4.47–24.05 | 8.77 | 5.79–17.13 |
| Mg (mg/L) | Urine | 2.04±1.30 | 1.01–5.68 | 1.22 | 1.04–4.87 | 4.54±2.04** | 1.06–9.11 | 4.42 | 1.52–8.49 |
| | Serum | 18.74±2.47 | 14.42–26.26 | 18.85 | 14.85–23.54 | 19.99±3.45* | 14.41–28.35 | 19.22 | 15.02–26.05 |
| Mo (µg/L) | Urine | 21.01±3.57 | 12.16–28.35 | 22.12 | 14.52–25.38 | 22.83±4.93* | 10.26–38.30 | 22.26 | 14.61–31.85 |
| | Serum | 5.44±1.03 | 3.24–7.99 | 5.36 | 4.03–7.55 | 6.28±1.40** | 4.16–8.89 | 6.09 | 4.27–8.81 |
| Ni (µg/L) | Urine | 18.51±4.00 | 10.24–26.79 | 18.12 | 11.31–25.10 | 44.01±27.37** | 10.01–97.53 | 37.80 | 10.49–97.22 |
| | Serum | 8.51±3.45 | 3.03–19.57 | 8.01 | 3.52–14.99 | 9.84±4.50 | 3.82–23.62 | 8.45 | 4.69–20.35 |
| K (mg/L) | Urine | 8.14±4.16 | 2.08–17.56 | 7.21 | 2.60–16.62 | 9.04±4.91 | 2.05–21.96 | 8.28 | 2.90–21.50 |
| | Serum | 145.38±28.88 | 90.18–189.33 | 148.10 | 99.07–184.82 | 144.23±27.66 | 93.55–183.43 | 156.70 | 96.76–177.02 |
| Rb (mg/L) | Urine | 212.87±46.47 | 140.90–315.10 | 204.83 | 145.94–306.40 | 220.06±41.58 | 144.50–323.00 | 218.25 | 155.55–304.30 |
| | Serum | 2.73±0.46 | 1.93–3.90 | 2.60 | 2.14–3.63 | 3.35±1.09** | 1.92–6.89 | 3.09 | 2.06–6.25 |
| Se (µg/L) | Urine | 3.39±0.70 | 1.72–5.77 | 3.32 | 2.22–4.62 | 2.73±0.49** | 1.56–3.99 | 2.80 | 1.59–3.60 |
| | Serum | 84.67±17.52 | 42.90–144.13 | 82.93 | 55.55–115.42 | 68.16±12.16** | 39.00–99.70 | 69.98 | 48.83–90.07 |
| Ag (µg/L) | Urine | 10.08±3.31 | 3.61–19.97 | 11.11 | 4.46–14.17 | 21.65±10.26** | 4.10–44.23 | 21.21 | 5.51–42.05 |
| | Serum | 0.05±0.02 | 0.002–0.13 | 0.05 | 0.01–0.09 | 0.06±0.03 | 0.003–0.17 | 0.05 | 0.01–0.14 |
| Na (mg/L) | Urine | 0.10±0.03 | 0.04–0.20 | 0.11 | 0.04–0.14 | 0.14±0.05** | 0.04–0.25 | 0.14 | 0.06–0.23 |
| | Serum | 339.19±26.38 | 291.01–394.97 | 338.02 | 293.98–382.21 | 346.72±34.46 | 289.57–398.97 | 351.77 | 291.02–392.62 |
| Sr (µg/L) | Urine | 924.07±185.46 | 609.10–1840.00 | 904.25 | 713.44–1163.80 | 901.07±130.01 | 717.56–1294.00 | 913.85 | 722.90–1107.83 |
| | Serum | 30.06±19.99 | 8.59–92.20 | 26.68 | 9.70–85.01 | 30.72±19.19 | 8.05–85.98 | 25.4 | 10.54–74.59 |
| Tl (µg/L) | Urine | 12.64±2.58 | 8.05–18.63 | 12.56 | 8.38–17.27 | 13.52±2.71 | 8.03–20.36 | 13.67 | 9.23–17.96 |
| | Serum | 0.11±0.07 | 0.01–0.33 | 0.11 | 0.02–0.23 | 0.15±0.10* | 0.02–0.43 | 0.13 | 0.04–0.40 |
| V (µg/L) | Urine | 0.13±0.02 | 0.09–0.20 | 0.12 | 0.09–0.17 | 0.09±0.02**** | 0.05–0.16 | 0.09 | 0.06–0.14 |
| | Serum | 31.56±4.05 | 19.5–42.73 | 31.65 | 24.54–38.65 | 40.10±8.07** | 26.90–60.83 | 39.23 | 28.03–59.26 |
| Zn (mg/L) | Urine | 21.00±2.68 | 15.54–25.90 | 21.54 | 16.04–24.65 | 23.77±3.81** | 16.70–32.81 | 23.89 | 17.63–31.50 |
| | Serum | 1.27±0.26 | 0.85–2.04 | 1.24 | 0.89–1.79 | 0.90±0.22** | 0.46–1.59 | 882.73 | 0.61–1.37 |
| | Urine | 982.47±54.77 | 904.66–1136.14 | 966.10 | 910.49–1108.40 | 687.62±156.77** | 442.20–999.76 | 689.80 | 465.56–983.30 |

Note

*p<0.05

** p<0.01 compared with controls.

Urine concentrations of 28 elements in workers and controls were summarized in Table 3. Twenty-two of the 28 (79%) elements showed significantly different concentrations in the urine of workers and controls. In comparison to control subjects, the data showed statistically significant increases in urine concentrations of In (295%), Al (17%), As (130%), Be (12%), Bi (15%), Cd (36%), Ca (18%), Cs (24%), Cr (128%), Co (321%), Cu (64%), Fe (75%), Li (35%), Mn (123%), Mg (9%), Mo (138%), Se (115%), Ag (40%), V (17%), and significant decreases in Rb (19%), Tl (31%), Zn (30%).

## Effect of employment years and concentrations of indium in serum and urine

A linear regression analysis was used to explore whether concentrations of indium in serum or urine changed as the function of worker's duration of exposure. The employment years of indium-exposed worker were further divided into ≤5 year, between 6 years and 14 years, and ≥15 years. Among the indium-exposed workers, significant increases in serum and urine indium concentrations were observed when the years of employment exceeded 6 years ($P < 0.05$) (Table 4). There was no statistically significant difference between the groups of 6–14 years of employment and ≥15 years of employment.

## Correlation analysis of elements in serum or urine

When serum concentrations of In, Al, As, Ba, Cd, Cs, Co, Se, Ag, and Zn were performed correlation analysis with urine concentrations of the same element, significant correlations were observed for In (r = 0.935, $P < 0.01$), Al (r = 0.602, $P < 0.01$), As (r = 0.499, $P < 0.01$), Ba (r = 0.699, $P < 0.01$), Cd (r = 0.476, $P < 0.01$), Cs (r = 0.869, $P < 0.01$), Co (r = 0.325, $P < 0.05$), Se (r = 0.441, $P < 0.01$), Ag (r = 0.317, $P < 0.05$), and Zn (r = 0.441, $P < 0.05$), but not for Be, Bi, Ca, Cr, Cu, Fe, Pb, Li, Mn, Mg, Mo, Ni, K, Rb, Na, Sr, Tl, V (Fig 2).

With regard to the degree to which element concentrations changed in serum and urine between indium-exposed workers and controls, the percentage of changes in serum In (696% increase vs. controls), Al (28%), Be (31%), V (27%), were larger than those in urine In (295%), Al (17%), Be (12%), V (17%); the changes of serum Cd (27%), Cr (55%), Cu (19%), Li (16%), Mn (35%), Mg (7%), Mo (15%) and Zn (29%), however, were smaller than those in urine Cd (36%), Cr (128%), Cu (64%), Li (35%), Mn (123%), Mg (9%), Mo (138%) and Zn (30%); the changes of Cs content in serum and urine were the same (24%); Se declined in serum(19%) but rose in urine(115%) in the indium-exposed workers compared with the controls; likewise Tl and Rb rose in serum(36%, 23%) but declined in urine(31%, 19%). Therefore, we further analyzed the correlation between changes of 15 elements (Al, Be, Cd, Cs, Cr, Cu, Li, Mn, Mg, Mo, V, Zn, Se, Tl and Rb) concentrations and indium in the same biological fluid (serum and

**Table 4. Serum indium and urine indium concentrations among workers with different length of service.**

| Group | n | Serum indium (µg/L) | Urine indium (ng/L) |
|---|---|---|---|
| | | Mean± SD (Range) | Mean± SD (Range) |
| ≤ 5 years | 30 | 30.61±12.31 (11.86–58.68) | 5.90±4.38 (0.07–13.90) |
| 6–14 years | 15 | 46.00±32.88 (17.60–137.63)* | 11.57±14.28 (0.72–54.93)* |
| ≥ 15 years | 12 | 52.45±33.14 (27.03–137.25)** | 14.67±12.71 (4.33–46.89)** |

Note

*$p<0.05$

** $p<0.01$ compared with workers of ≤5 years.

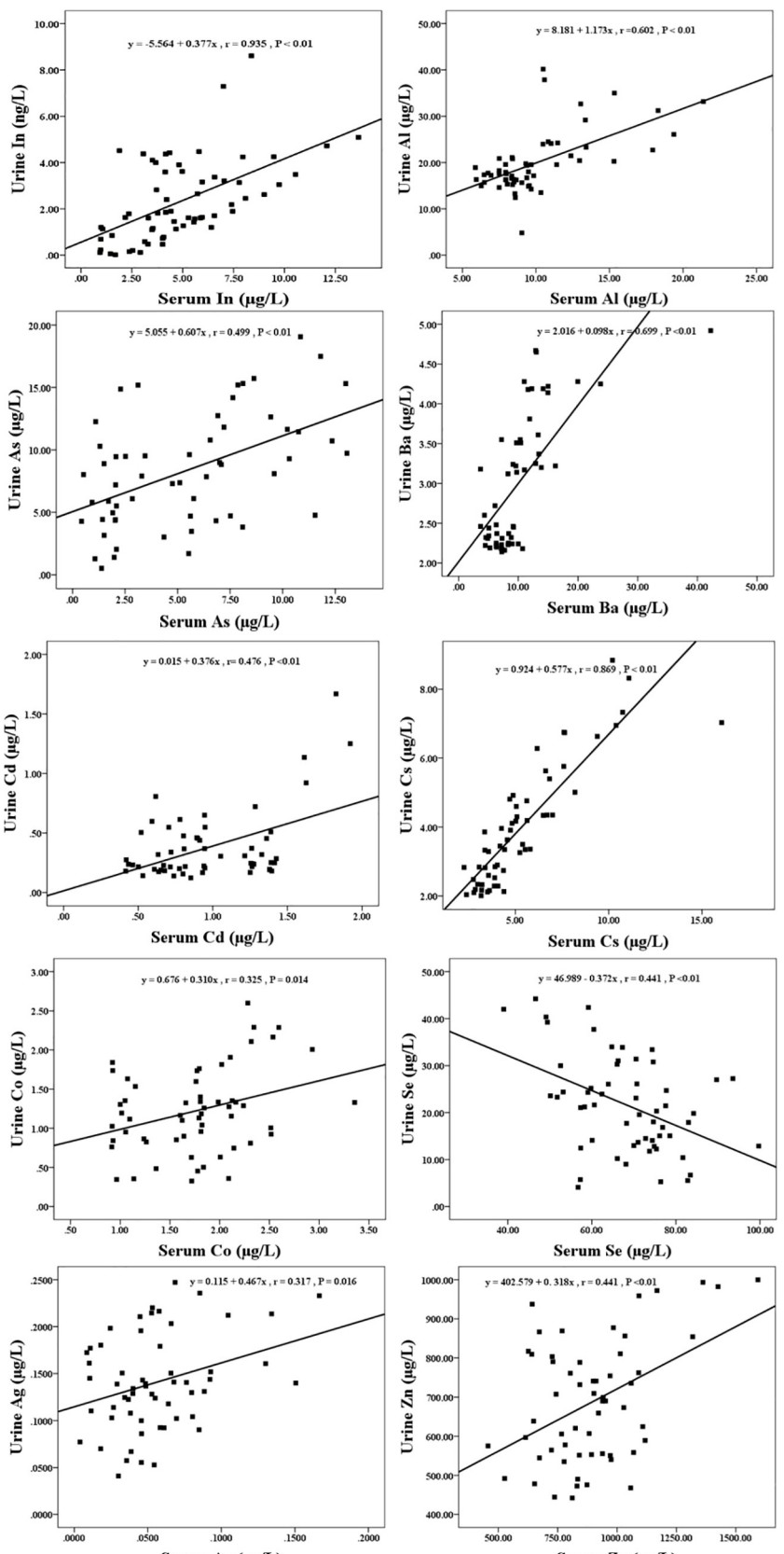

**Fig 2. Correlation between serum indium and urine indium among 57 workers.** Data were analyzed by linear correlation with 95% confidence interval, with a correlation coefficient (r) = 0.935 and $P < 0.01$.

urine) (Table 5). There were significant positive correlations between In and Be, Cd, Cr, Zn in serum as well as in urine. There was significant positive association between In and Se in urine, however, In concentrations were inversely associated with those of Se in serum. There were no significant correlations between In and Cs, Cu, Li, Mn, Mg, Mo, Tl, Rb in either serum or urine.

Table 6 shows multiple linear regression analyses for indium concentration (log transformed) in serum and urine adjusted for age, length of service, gender, smoking and alcohol consumption, all elements concentration in serum and urine.

## Discussion

The main source of indium in the working environment is the manufacturing process of melting raw materials which releases indium to the atmosphere around the furnace as industrial dust [11, 12]. The data presented in this report clearly show that indium was emitted in the fume produced during the production process. The highest concentration of airborne indium surrounding the workers' breathing zones inside the workshop exceeded the permissible concentration-time weighted average (PC-TWA, 0.1 mg/m$^3$) by more than eleven times. Cynthia's group in USA reported a similar finding [13]. By using an air sampler with a 37-mm, 0.8-μm MCE filters, they found that the airborne indium level was about 0.072 mg/m$^3$ during cleaning an indium-tin oxide (ITO) evaporation chamber. Kristin *et al.* have also reported that air indium concentrations varied from 0.0004 to 0.108 mg/m$^3$ during the production process [14]. The values in our study appeared to be higher than that detected by Kristin's team. The mean value of airborne indium in our study was 0.078 mg/m$^3$, and ranged from 0.001 to 1.12 mg/m$^3$. The results are consistent with Ichiro's observations (AM indium concentration was 0.095 mg indium/m$^3$ as total dust, and ranged from 0.00040 to 1.27 mg In/m$^3$) and support the occupational risk of workers exposed to airborne indium [15]. Owing to respiratory tract is the main way of workers exposed to indium and its compounds, it is necessary to adopt perfect ventilation and dust removal facilities in the workplace. At the same time, it is a duty to improve the awareness of occupational health protection of enterprise manager and workers, and scientifically to prevent the occupational hazards of indium and its compounds.

At present, some researchers believe that the concentration of indium in serum or urine can be used as a biological indicator or biomarker of indium exposure [16–19]. The indium-

**Table 5. Correlation coefficients between indium and other elements in serum and urine of indium-exposed workers.**

|  | Serum | Urine |
|---|---|---|
| Al—In | $r = 0.398$ ($P > 0.05$) | $r = 0.436$ ($P < 0.01$) |
| Be—In | $r = 0.406$ ($P < 0.01$) | $r = 0.425$ ($P < 0.01$) |
| Cd—In | $r = 0.473$ ($P < 0.01$) | $r = 0.626$ ($P < 0.01$) |
| Cr—In | $r = 0.582$ ($P < 0.01$) | $r = 0.456$ ($P < 0.01$) |
| Se—In | $r = -0.593$ ($P < 0.01$) | $r = 0.326$ ($P < 0.05$) |
| V—In | $r = 0.331$ ($P < 0.05$) | $r = 0.032$ ($P > 0.05$) |
| Zn—In | $r = 0.735$ ($P < 0.01$) | $r = 0.547$ ($P < 0.01$) |

Note: Data from In-exposed workers were combined and analyzed by linear regression.

**Table 6. Forward stepwise multiple linear regression analysis of elements concentration (logtransformed) in serum and urine studied.**

| | Biological sample | Adj R² | Predictor | B | P | Partial correlation |
|---|---|---|---|---|---|---|
| Indium | serum | 0.532 | As | 0.296 | 0.005 | 0.369 |
| | | | Be | 0.225 | 0.017 | 0.319 |
| | | | Cd | 0.245 | 0.011 | 0.338 |
| | | | Cr | 0.277 | 0.008 | 0.350 |
| | | | Zn | 0.087 | 0.000 | 0.735 |
| | urine | 0.381 | age | 0.250 | 0.018 | 0.323 |
| | | | As | 0.242 | 0.031 | 0.289 |
| | | | Cd | 21.870 | 0.000 | 0.626 |
| | | | Cs | -1.844 | 0.002 | -0.409 |
| | | | Mg | -0.402 | 0.040 | -0.283 |
| | | | Zn | 0.355 | 0.001 | 0.418 |

Note: The following variables entered into the models: age; length of service; gender (0: male; 1: female); smoking (1: yes; 0: no); alcohol consumption (0: no; 1; yes); only indium and elements values with p-values that are significant are presented; all elements concentration in serum and urine.

exposed workers did show a significantly higher indium level in serum or urine as compared to control subjects [20]. Makiko has summarized that heavy indium exposure is a risk factor for emphysema, which can lead to a severity level, such as emphysematous changes have been highlighted as a long-term adverse effect on lungs in indium-exposed individuals with serum indium 20 μg/L in their 5-year follow-up study as well as a separate 8-year follow-up study, even after adjusting for age, duration since initial indium exposure, and smoking history [21]. A recent epidemiological study including 87 cases of ITO workers further suggests that plasma indium concentration reflected cumulative respirable indium exposure, which was associated with clinical, functional, and serum biomarkers of lung disease [14]. Serum or urine indium levels may reasonably indicate the status of systemic indium following recent exposure among the active indium-exposed workers. Therefore, this study used serum and urine samples as a vivo biomonitoring of indium and its compounds. The results from this study showed that long-term, chronic exposure to indium and its compounds had a significantly impact on level of serum and urine indium. Serum and urine concentrations of indium among workers were about 8-fold and 4-fold greater than those detected in controls. The increases in levels of serum and urine indium appeared to be related exposure time (Table 3). When the employment years of indium-exposed worker were stratified into three groups, compared with the group of ≤5 employment years, significant increases in serum indium and urine indium concentrations were observed in the group with 6–14 years and ≥15 years of employment. It is suggested that the increase of work-length could cause the increase levels of indium in serum and urine.

Another finding is the fact that exposure to indium and its compounds also had effects on serum concentration of As, Ba, Be, Cd, Cr, Se, V, and Zn, i.e., an increase in serum As, Ba, Be, Cd, Cr, V, and a decrease in serum Se, Zn. The same is true for urine concentrations of Al, As, Be, Cd, Cr, Se, and Zn, i.e., an increase in urine Al, As, Be, Cd, Cr, Se, and a decrease in urine Zn. Exposure to indium compounds resulted in altered levels of In, Al, Be, Cd, Cs, Cr, Li, Mn, Mg, Mo, V, Se and Zn in serum and urine of indium exposed workers. A comprehensive analysis of the changes in the concentration of these elements in serum and urine, it appeared that element concentrations, particularly Be, Cd, Cr, Se, and Zn, are significantly association with the serum indium and urine indium. This association could partly be attributed to the possible action of indium on trace elements metabolism. However, the fact that the indium production

process contains more metals including lead, arsenic, cadmium, zinc suggests that the elevated serum and urine concentrations of these elements may be a direct result of the overexposure to these metals in the fume. It is also possible that differences in toxicokinetics among different years of employment groups may contribute to higher serum levels of these metals. Perhaps the higher exposure in the indium exposed workers will really make this group of workers more vulnerable to metal toxicities, after all, our previous animal experiments have found that the trace elements imbalance in rats exposed to ITO exposure rats [22].

Our results demonstrate that measurement of elements concentrations in both serum and paired urine rather than only in serum or urine are important to get insights into the roles of these elements in disease. The results showed that serum indium concentrations were significantly correlated with serum Be, Cd, Cr, Se, Zn, respectively; the same was true for urine indium and urine Be, Cd, Cr, Se, Zn. Among metals of health concern, Se and Zn are essential trace elements and involved in numerous biochemical processes that support life, but yet their deficiency or overload is detrimental to human health [23]. Results of the present study indicate that workers chronically exposed to indium may decrease serum Se and Zn level compared to the control group. Since there is no homeostatic control of indium in the human body, it is toxic trace element. We speculate that when high amounts of indium are introduced into the body, it may affect the normal metabolism of Zn or Se in human body, and then may replace Zn or Se at the key enzyme sites causing metabolic disorders, and eventually lead to a significant decrease in serum Zn or Se levels in workers chronically exposed to indium. The previous research of our group shows that following chronic exposure of ITO to rats, the Zn content decreases in lung and the Cu content increases in lung. Since both Cu and Zn are actively participating in cellular redox reactions, changes in the homeostasis of these two metals in serum and urine of indium-exposed workers may aggravate the damage of reactive oxygen species to cells, and ultimately contribute to the pulmonary toxicity caused by indium and its compounds [24, 25]. The causal correlation between the element (Se and Zn, for example) content and disease is complicated, and so far we did not know why concentrations of Se and Zn were perturbed in indium-exposed workers. But the low levels of Se and Zn in the indium occupational workers are alarming. Trace metals such as As, Be, Cd and Cr, on the other hand, are toxic metals with no beneficial health effects and have been recognized as human or animal carcinogens by International Agency for Research on Cancer [26–28]. The effect of occupational exposure to indium on the metabolism of toxic metals appears to be poorly understood. However, results of the present study indicate that exposure to indium may influence serum and urine levels of As, Be, Cd and Cr. Therefore, their interactions with indium may affect various fundamental biological processes, including intra- and intercellular signaling, apoptosis, and ionic transportation. Besides, we don't known whether toxic metals could influence indium's absorption, distribution, deposition, and excretion processes.

Taken together, we believe that element analysis from serum and urine provide an additional means for assessment of indium concentrations in indium occupational workers. Trace metal elements play an important role in biological processes by coordinating enzymatic reactions or by affecting the permeability of cell membranes, among others. Since exposure to indium and its compounds apparently alters the homeostasis of these trace elements, future studies should investigate the importance of these trace metals in indium-associated pulmonary toxicity.

## Limitations of the study results

This study has limitations without doubt. First, a larger sample size and more kinds of biological sample are desirable for a more accurate estimation, the impact of random errors might be

large owing to the small sample size. Second, false positive rate may have resulted owing to the multiple statistical testing problem. Adjustment of p-values (and confidence intervals) is usually unrealistic when large numbers of statistical tests are performed in an exploratory analysis. In such situations, it is difficult to accurately quantify the total number of tests performed and their interconnectedness, and the adjusted threshold for statistical significance is very small (and the false-negative rate is very high) because of the large number of tests performed. Therefore, it is important to interpret results cautiously rather than trying to accurately determine the true significance level of the p-value. Third, the blood indium level of local residents was unknown. In this study, the control subjects would have measurable indium in serum and urine. The population in our research is in Guangxi, which is the largest indium production base in China, and there are many local production and processing enterprises. Although there was no occupational indium exposure in the control subjects, the indium pollution in river or soil was not ruled out. Meanwhile, because the background of indium exposure was similar for indium-exposed workers and controls, it was likely to have no or minimal impact on the analysis. Our future studies will be directed to overcome these limitations.

## Generalization of the study results

Because only factories and workers from one region of China were included in this study, the scope of the results may be limited by the geographical characteristics. Furthermore, because data were collected cross-sectionally, the change of trace elements in urine can only reflect the recent exposure, we can't infer the influence of diet and other changes on the elements in urine. Therefore, these results should be generalized only after careful consideration.

## Conclusions

To the best of our knowledge, the current research represents the first scientific contribution reporting levels of 28 elements in two biological samples across indium-exposed population. The results of this study provide the information regarding serum and urine concentrations of elements in workers of indium ingot production in China. The serum and urine levels of some elements are modified by exposure to indium in occupationally exposed workers. We showed that the concentrations of 13 elements were increased or decreased in serum and urine compared to control groups, suggesting that the abnormal expression of these elements may have a potential effect on the lung toxicity induced by indium and its compounds. These findings shall be useful for future research to monitoring occupational and environmental exposure and to compare the indium exposure levels within China and around the world.

## Supporting information

**S1 Data.**
(SAV)

**S1 Questionnaire.**
(DOC)

## Acknowledgments

The authors gratefully acknowledge the technical assistance of Cheng Juan who is in Institute for Occupational Health and Poison Control under the China Center for Disease Control and Prevention (CDC).

## Author Contributions

**Data curation:** Nan Liu.

**Formal analysis:** Nan Liu.

**Investigation:** Nan Liu, Yi Guan.

**Methodology:** Nan Liu.

**Project administration:** Sanqiao Yao.

**Resources:** Bin Li.

**Supervision:** Sanqiao Yao.

**Writing – original draft:** Nan Liu.

**Writing – review & editing:** Nan Liu.

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
