## [Decision Letter · Decision Letter 0]

4 Aug 2020

PONE-D-20-20542

Biomonitorization of Concentrations of 28 Elements in Serum and Urine among Workers Exposed to Indium Compounds

PLOS ONE

Dear Dr. yao,

Thank you for submitting your manuscript to PLOS ONE. After careful consideration, we feel that it has merit but does not fully meet PLOS ONE’s publication criteria as it currently stands. Therefore, we invite you to submit a revised version of the manuscript that addresses the points raised during the review process.

We look forward to receiving your revised manuscript.

Kind regards,

Yi Hu

Academic Editor

PLOS ONE

Journal Requirements:

- https://www.sciencedirect.com/science/article/abs/pii/S0378427407009757?via%3Dihub

- https://www.ncbi.nlm.nih.gov/pmc/articles/PMC4126160/

- https://link.springer.com/article/10.1007/s13205-012-0072-6

-https://www.sciencedirect.com/science/article/abs/pii/S1352231015002356?via%3Dihub

In your revision ensure you cite all your sources (including your own works), and quote or rephrase any duplicated text outside the methods section. Further consideration is dependent on these concerns being addressed.

Reviewers' comments:

Reviewer's Responses to Questions

**Comments to the Author**

1. Is the manuscript technically sound, and do the data support the conclusions?

Reviewer #1: Yes

Reviewer #2: Partly

Reviewer #3: Yes

2. Has the statistical analysis been performed appropriately and rigorously? 

Reviewer #1: Yes

Reviewer #2: No

Reviewer #3: Yes

3. Have the authors made all data underlying the findings in their manuscript fully available?

Reviewer #1: Yes

Reviewer #2: No

Reviewer #3: Yes

4. Is the manuscript presented in an intelligible fashion and written in standard English?

Reviewer #1: Yes

Reviewer #2: Yes

Reviewer #3: Yes

5. Review Comments to the Author

Reviewer #1: This manuscript measures indium and 27 other elements/metals via mass spec. in workers at an indium ingot production facility and includes a comparison/control group of workers at a nearby facility in China (industry for comparison group not identified – but considered unexposed to indium). Area and personal air sampling were also performed at the indium ingot production facility. For the indium-exposed group, serum and urine indium was stratified by tenure. All elements/metals were compared in the serum and urine of indium-exposed workers and the comparison group. Correlations between serum and urine levels were measured for all elements/metals. The four components of the study are outlined below.

• Mean indium measured from air sampling was 78 μg/L.

• Indium in serum and urine increased dramatically by tenure.

• Some of the 27 elements/metals measured were higher in serum and urine in the indium-exposed workers compared to the control group (Aluminum, Beryllium, Cadmium, Chromium, Copper, Lithium, Magnesium, and Molybdenum), Zinc was lower in serum and urine in the indium-exposed workers than the control, and other elements/metals were higher in one specimen (serum or urine) but lower in the other in the indium-exposed workers (serum Selenium lower in indium-exposed workers, urine Selenium higher in indium-exposed workers; serum Thallium and Rubidium higher in indium-exposed workers than control group, urine Thallium and Rubidium lower in indium-exposed workers than control group).

• A number of elements had positive correlations with serum and urine levels, selenium had a negative correlation, and other elements’ serum and urine levels did not correlate.

The manuscript is well written and warrants publication. Consider a follow up health study in the future if that is a possibility. A few additional considerations below:

Abstract:

When listing elements in Results, be sure to specify serum, urine, or both.

Methods:

Demographics are more appropriate in the Results section

‘Were non-alcohol drinkers’ > ‘did not drink alcohol’

Discussion:

Kristin et al. > Cummings et al.

Still should address how controls could have that level of serum indium? Environmental contamination from proximity to indium facility?

Can you include any thoughts on why Thallium and Rubidium would be higher in blood but lower in urine in the indium-exposed workers compared with the controls? Likewise Selenium lower in blood but higher in urine?

‘At present, it is uncertain whether the concentration of indium in serum or urine can be used as a biological indicator or biomarkers of indium exposure’ I don’t think this is true, I would say they can be used as biological indicators of exposure?

Reviewer #2: [Overall Comment]

Thank you for the opportunity to review the article "Biomonitorization of Concentrations of 28 Elements in Serum and Urine among Workers Exposed to Indium Compounes (PONE-D-20-20542)". Since indium-exposed worker has become important health issue, an effective prevention is an urgent matter. Exposure will need to be prevented before health problems appear. It would also be helpful to know the biokinetics of metals. This study will provide important insights into the prevention of these metal-related health problems. However, there are a few concerns with this paper. I hope the authors will consider the matters I have listed.

1. Some paragraph in the Introduction section and the Discussion section is not paragraph writing. The authors should try to write one topic per one paragraph.

2. There seems to be a paucity of cited references. Please do a more careful review of previous studies.

3. A large number of statistical tests have been performed. Therefore, it is necessary to control for the family-wise error rate. There are 120 subjects in the analysis and 133 p-values in the tables and charts. This do clearly cause errors.

4. The Discussion section is very long and redundant. It needs to be more concise.

5. Please check the STROBE statement. Then, please check your manuscript using the STROBE checklist.

6. Please check your English grammar.

[Comment]

I show the page and line number of the Word file (manuscript.docx), "quote from the text", and my comments.

7. Page 1, Line 16, "27 elements": I think "27" is wrong. Is it "28 elements" ?

8. Page 1, Line 23, "Ambient indium levels ... 78.41 μg/m3.": The Abstract-Method section don't contains the method that the ambient indium level measure. Please add the method.

9. Page 1, Lines 25-26, "the variation in ... among study populations.": there are no results of analysis to support the association between serum/urine indium level and airborne indium level. Please provide the results of the analysis that support this.

10. Page 2, Line 33, "Linear regression analysis": Is the "r" you are presenting here a standardized regression coefficient ? If so, please describe that.

11. Page 3, Lines 77-79, "we have validated ... (ICP-MS)." : Please cite the authors' validated results, if they report them.

12. Page 3, Lines 81-84, "To the best ... indium-exposed population.": I believe this should be included in the Discussion section.

13. Page 3, Line 85, "toxic metals": The indium-exposed worker be exposed to any other toxic metal, too? If so, what other metals besides indium will the indium-exposed worker be exposed to? Please describe that.

14. Page 4, Lines 90-92, "Hence this study ... its compounds exposure.": I believe this should be included in the Discussion section.

15. Page 4, Lines 100-115: I believe some of this part should be included in the Result section.

16. Page 4, Line 100: Did the exposed workers wear personal protective equipment? Please describe.

17. Page 4, Lines 107-110, "The control subjects ... and other metals.": Please describe more details on how the control subjects were recruited. Are they volunteers? Or were they selected at random? Or were they selected by the researcher to be similar to the exposed workers?

18. Page 4, Lines 113-114, "The two groups ... no significace differences": No statistical test at baseline is required. Please check the STROBE statement.

19. Page 5, Line 118 & Line 123-124: Similar overlap. Please merge them together.

20. Page 5, Line 124-125: Please include approval number.

21. Page 6, Line 148, "~10%": This is 15%, isn't it? (0.045 μg/0.0003 mg = 0.000045 mg/0.0003 mg = 0.15). If so, please replace "~10%" with "15%".

22. Page 6, Line 151, "local clinics": Please provide a little more detail about "local clinics".

23. Page 6, Line 155-156, "Nan et al, 2017": Incorrect citation style.

24. Page 6, Line 156 "℃until": Insert a space between "C" and "u", please.

25. Pages 6-7, Lines 172-184: Please put them in a table, they are difficult to read in the main text.

26. Page 7, Line 188, "Student's t test": Since the variance is unknown, it is Welch's t test is better than Student's t test.

27. Page 7, Line 188, "paired samples": In the present study, I believe the subjects were not paired. If indium-exposed workers and control workers were paired, please describe how did the authors make paired subjects.

28. Page 7, Lines 189-190, "ANOVA":In the Result section, I couldn't find the results of the ANOVA. Please describe more clearly the method and result of ANOVA.

29. Page 7, 191-192, "by a linear regression": A multiple regression analysis has also been done, but it is not listed in Method section. Please stated it. And, please describe an explanation that the standardized regression coefficients were calculated.

30. Page 7, Lines 196-197, "No biological sample ... level of indium.": The first sentence of paragraph is the sentence of the paragraph's topic. In this paragraph, the first sentence should be a brief description of the airborne indium level. Please rewrite the paragraph to make it more clear.

Reviewer #3: This study provides valuable data on indium concentrations from indium-exposed workers’ serum and urine. The authors also point out that indium affects different trace elements’ homeostasis in the systemic circulation. The paper is well organized and written. However, minor revision is needed, and the following concerns need to be addressed before the paper can be accepted for publication.

1. The author should elaborate on why serum and urine samples were selected for in vivo biomonitoring of indium and its compounds. What are the limitations of using only fluid samples as bio-monitor? Why not use blood or hair samples as a bio-monitor for those metal measurements?

2. Please include a paragraph about the quality assurance of your results.

3. p.16; It might be better to use the same units, such as µg/m3 (or mg/m3), that are consistent with your data and the critical literature concentrations.

4. “When high amounts of indium are introduced into the body, it maybe replace Zn or Se at the key enzyme sites causing metabolic disorders.” The author should provide more information for supporting this statement. Which physical or chemical-depended mechanisms might affect the decrease of serum Zn or Se under indium exposed?

5. “When indium-exposed worker’s years of employment were stratified into three employment groups, serum indium and urine indium concentrations in the group with 6-14 years and ≥15 years of employment were significantly higher than those with less than five employment years, suggesting a work-length related increase in serum and urine indium.” Could the authors conclude that indium’s residence time in vivo is quite long that cause indium more hazard to human health?

6. A fine-tuning in English is required.

6. PLOS authors have the option to publish the peer review history of their article (what does this mean?). If published, this will include your full peer review and any attached files.

Reviewer #1: No

Reviewer #2: **Yes: **Toshiharu Mitsuhashi

Reviewer #3: No

---

## [Author Response · Author response to Decision Letter 0]

13 Sep 2020

Reply to revision

Responses to Editor:

Thank you for the opportunity to revise my paper. An explanation of the additional requirements below:

1.I read the PLOS ONE style templates carefully, and ensure that my manuscript meets PLOS ONE's style requirements.

2.Thank you very much for pointing out some problems. In my revision, I cite all my sources (including my own works), and make certain appropriate readjustments to some texts. But the author still wants to explain the difference between our paper and the following previous publications.

Publication 1：Title: Alteration of saliva and serum concentrations of manganese, copper, zinc, cadmium and lead among career welders. The differences between our paper and the publication are research objects (indium-exposed workers are different from career welders), biological samples (serum and urine rather than saliva), and analysis elements (28 elements are far more than 5 elements).

Publication 2：Title: Occupational Exposure to Welding Fume among Welders: Alterations of Manganese, Iron, Zinc, Copper, and Lead in Body Fluids and the Oxidative Stress Status. The differences between our paper and the publication are research objects (indium-exposed workers are different from welders), the treatment method of biological samples (the digestion solution direct dilution method is simpler and more accurate than microwave digestion), and analysis elements (28 elements are far more than 5 elements).

Publication 3：Title: Trace elements and carcinogenicity: a subject in review. The differences between our paper and the publication are article type (research article is completely different from review article in essence), main idea of the article (one is to study the detection and analysis of 28 elements in indium-exposed workers, the other is to study the relationship between trace elements and carcinogenicity).

Publication 4：Title: Inhibition of the WNT/β-catenin pathway by fine particulate matter in haze: Roles of metals and polycyclic aromatic hydrocarbons. The author has never read this publication. Moreover, through the abstract of that article, I found that the research content of that article is metal and polycyclic aromatic hydrocarbons in haze. The detection objects of our paper are indium-exposed workers, which is essentially different from that article.

3.I have attached a questionnaire as part of this study. I upload this as a separate file labeled 'Questionnaire'. 

4.The corresponding author has an ORCID iD and has completed the Authorization process and have been logged in to Editorial Manager. 

Response to Reviewers:

Response Reviewer #1: 

Abstract:

When listing the elements in the results, the author has carefully revised the specific samples of each element, such as serum, urine or both.

Methods:

Demographics have been changed to the Results section.

In our study, individuals who had consumed alcohol more than twice a week during the previous 6 months were defined as drinkers.

Discussion:

In order to control the workers' serum indium level, we should control the environmental contamination from indium facilities and do a good job of workers' personal protection. Therefore, the author address solutions at the end of the first paragraph of the discussion.

Our results show that, some elements/metals were higher in one specimen (serum or urine) but lower in the other in indium-exposed workers, this increase or decrease could partly be attributed to the possible action of indium on trace elements metabolism. Perhaps the higher exposure in the indium exposed workers will really make this group of workers more vulnerable to metal toxicities. 

‘At present, it is uncertain whether the concentration of indium in serum or urine can be used as a biological indicator or biomarkers of indium exposure’: this statement is not correct, the author has corrected this sentence in the discussion section.

Response Reviewer #2: 

1.The author has made some modifications to some paragraphs in the Introduction section and the Discussion section.

2.By carefully reviewing previous studies, the author increases the number of cited references.

3.I'm very sorry, I didn't find the error you said. Please point it out.

4.The author has made some concise modifications to the discussion part.

5.The author submitted a Research Checklist. 

6.Thank you for the opportunity to check my English grammar.

7.This study was aimed to assess inddium and other 27 elements in serum and urine. The total is 28 elements.

8.The author has added the method of the ambient indium level measure in the Abstract-Method section.

9. "the variation in ... among study populations.": this conclusion is inaccurate and has been deleted.

10."Linear regression analysis": the "r" was a standardized regression coefficient. The standardized regression coefficient refers to the regression coefficient calculated after the data is standardized (minus the mean value and dividing the variance). It can eliminate the influence of dimension and order of magnitude after the standardized data, so that different variables are comparable. Therefore, the standardized regression coefficient can be used to compare the effect of different independent variables on dependent variables.

11."we have validated ... (ICP-MS)." : in this part of the content, the author's expression may be wrong, there is no “validated results” here. The author has made corrections in the paper. 

12. "To the best ... indium-exposed population.": this part has been changed to the Discussion section.

13."toxic metals": the indium-exposed workers were exposed to other toxic metal, too. It can be seen from the first paragraph of the Introduction section, “In the process of indium smelting, there are occupational hazards such as dust, lead, arsenic, cadmium, indium, zinc, hydrogen arsenide, various acids and bases, noise, high temperature.”.

14."Hence this study ... its compounds exposure.": This sentence is repeated with the Discussion section and has been deleted.

15.Demography has been placed in the Results section.

16.During operation, the exposed workers wear dust-free clothes, while maintenance workers wear full face filter cotton, goggles and protective gloves.

17.The control subjects were office workers from another factory who were not exposed to occupational hazards. They were volunteers, and were matched with the exposed workers for age, sex, and smoking habits.

18.General characteristics of the 120 study populations in the control and indium exposure group are summarized in Table 1. The distribution of age and frequencies of sex, smoking and drinking were not significantly different among the two groups.

Table 1 General characteristics and lifestyle habits in study populations.

Variable Controls Wokers statistics P

Number 63 57 

Age (year,±s) 39.32 ± 11.37 37.82 ± 9.34 0.77 0.44a

Sex (male/female,%) 46/17(73) 41/16(72) 0.02 0.89b

Smoking (yes/no, %) 31/32(49) 21/36(37) 1.86 0.17c

Drinking (yes/no, %) 20/43(32) 18/39(32) 0.00 0.98c

Years of exposure (year,±s) 8.97 ± 8.39 8.31 ± 7.54 0.45 0.65a

a Ln-transformed for statistical inference. b One-way ANOVA. c χ2 test.

19.The author has merged the overlap together.

20.The author has added the approval number.

21.Yes, it is 15%, the author has corrected the mistake.

22.The “local clinics” refers to the local occupational disease prevention and control hospital where the indium factory is located.

23.The author has corrected the incorrect citation style.

24.The author has inserted a space between "℃" and "u".

25.The author has put the information of 28 elements in Table 1.

26.The author has used Welch's t test to analyze the data.

27.The author has changed the description of the error.

28.The differences between two means were analyzed by independent samples T test.

29.A multiple regression analysis has listed in Method section,and stated it. 

30.The author has rewrited the paragraph to brief description of the airborne indium level.

Response Reviewer #3: 

1.The author has explained in the second paragraph of the Discussion section why serum and urine samples were selected for in vivo biomonitoring of indium and its compounds.

2.The author has attached a paragraph about the quality assurance of our results in the Method section.

3.The author has changed the unit of the airborne indium level into the same units, such as mg / m3.

4.As requested, the author has elaborated on this statement in the fifth paragraph of the Discussion section.

5.Through these research results, the author couldnot conclude that indium’s residence time in vivo is quite long that cause indium more hazard to human health. The author can only prove that the longer the working years of workers, the longer the exposure time of indium, resulting in the higher content of indium in serum or urine, which may cause more hazard to human health; however, this does not mean that the long accumulation time of indium in the body causes harm to human body.

6.I had readjusted my English grammar. If you feel there are any problems, please let me know as soon as possible, and I will try my best to adjust it.

---

## [Decision Letter · Decision Letter 1]

7 Oct 2020

PONE-D-20-20542R1

Biomonitorization of Concentrations of 28 Elements in Serum and Urine among Workers Exposed to Indium Compounds

PLOS ONE

Dear Dr. yao,

Thank you for submitting your manuscript to PLOS ONE. While most reviewers' comments have been addressed,  I would ask the authors to address the issue about the measured indium levels in the control subjects - it suggests background indium exposure or previous occupational exposure, but indium is not a trace element and we don't expect dietary exposure, but at the same time, there have not been large studies to measure blood indium in unexposed populations. Still, it is worth a sentence or two in the discussion. 

We invite you to submit a revised version of the manuscript that addresses the points raised during the review process. Please submit your revised manuscript by Nov 21 2020 11:59PM. If you will need more time than this to complete your revisions, please reply to this message or contact the journal office at plosone@plos.org. Please include the following items when submitting your revised manuscript:

We look forward to receiving your revised manuscript.

Kind regards,

Yi Hu

Academic Editor

PLOS ONE

Reviewers' comments:

Reviewer's Responses to Questions

**Comments to the Author**

1. If the authors have adequately addressed your comments raised in a previous round of review and you feel that this manuscript is now acceptable for publication, you may indicate that here to bypass the “Comments to the Author” section, enter your conflict of interest statement in the “Confidential to Editor” section, and submit your "Accept" recommendation.

Reviewer #1: All comments have been addressed

Reviewer #2: (No Response)

2. Is the manuscript technically sound, and do the data support the conclusions?

Reviewer #1: Yes

Reviewer #2: Yes

3. Has the statistical analysis been performed appropriately and rigorously? 

Reviewer #1: Yes

Reviewer #2: No

4. Have the authors made all data underlying the findings in their manuscript fully available?

Reviewer #1: Yes

Reviewer #2: Yes

5. Is the manuscript presented in an intelligible fashion and written in standard English?

Reviewer #1: Yes

Reviewer #2: No

6. Review Comments to the Author

Reviewer #1: The manuscript is much improved and most reviewer comments were addressed. The lingering question I think should be addressed is why the control subjects would have measurable indium in blood?

Reviewer #2: [Overall comment]

Thank you for revising the manuscript. Revisions have been made in a number of places and I believe that the research paper becomes more qualified for the readers.

However, I still have concerns about some points I have ever pointed. There are also some new concerns that should be pointed out in the revised manuscript. Please review and revise the manuscript.

[Comment continued from last time; comment number x-r1]

*3-r1. multiplicity of statistical tests

I apologize for my earlier point, which was poorly worded. I will be more specific below.

In a statistical test, there is a 5% chance of p<0.05, even for comparisons that are not truly different (i.e., Type I error). So, for example, if there are 50 reported p-values, the probability that one or more of them will be p<0.05 by only chance is about 92%.

In this study, more than 100 p-values were calculated (for example, although not reported, I think we calculated p-values even for metals that are not marked with "*" or "**" in Table 2. ). With such a large number of calculated p-values, it is impossible to distinguish between cases where there is a truly significant difference and cases where the difference appears to be significant by type I errors.

In this study, both cases of p-values are considered to be mixed. In other words, some of the comparisons that resulted in p<0.05 in this study are just coincidental.

Please make the following two points on this issue

a) Please address it statistically.

b) Please consider this issue in the Discussion section.

Reference: the number of p-values

Table 2=56, Table 3=4, Table 4=54, Table 5=11, Figure 1=10. that is to say, there are at least 135 p-values in this manuscript.

*5-r1. Check List

Thank you for completing and submitting the Check List (SRQR guideline for Qualitative research). However, this study is not a qualitative research, so this checklist is not appropriate. Therefore, I believe that this checklist is inappropriate.

I think it would be better to use STROBE (https://www.strobe-statement.org/index.php?id=strobe-home).

*10-r1. standardized regression coefficient

Thank you for describing r. I've always known what standardized regression coefficient is. What I wanted to point out in comment #10 is that it was not stated in the Method section that "standardized regression coefficients are reported in linear regression analysis".

Please add and explain this description in the Method section.

*16-r1. personal protective equipment

Thank you for describing your personal protective equipment.

Please explain this in the Method section (Subject) as well as in the rebuttal letter.

*17-r1. Control subject

Thank you for explaining the controls (non-exposed group). Please explain this information in the body of the manuscript as well as in the rebuttal letter.

You mentioned that you matched workers at the indium plant (exposed group) and controls (non-exposed group) with age, gender and smoking history, but the number of person is 57:63. Why is this? Please add an explanation to the main text.

And, there is a difference in the percentage of smokers between the groups, even though they were matched. Please add an explanation for this in the main text.

*18-r1 Comparison between workers and controls

Table 1 in rebuttal letter is also required for the manuscript, so please add it.

However, I believe that statistical tests should be removed. If the controls is matched to the workers, there must be no significant difference in age, gender, and smoking habits, which are not required to be tested.

And, the significant test in baseline demographic data is not recommended in the STROBE statement (JP Vandenbroucke et al, 2007, https://pubmed.ncbi.nlm.nih.gov/18049195/). The following is a quote from JP Vandenbroucke 2007, p.822.

"Inferential measures such as standard errors and confidence intervals should not be used to describe the variability of characteristics, and significance tests should be avoided in descriptive tables. Also, P values are not an appropriate criterion for selecting which confounders to adjust for in analysis; even small differences in a confounder that has a strong effect on the outcome can be important."

[New comments]

31. Limitation

Please add the limitations of this study in the Discussion section.

32. Duplicate expressions.

The following are duplicates, please merge them.

Page 6, Lines 162-163: "P values less than 0.05 indicate statistical significance."

Page 6, Lines 169-170: "All statistical tests are two-sided with a significance level of 0.05."

33. emotional expression

It is best to avoid expressing emotions in academic papers.

Page 11, Line 292: "More surprising"

Page 12, Line 329: "Noticeably"

34. Description to be included in the Result section

Page 11, Line 312 - Page 12, Line 326: "It was also found that ... Se (115%) and Zn (30%). "

Much of this paragraph is the description of results and should not be in the Discussion section. Please minimize the description of results in the Discussion section and put the description of results in the Results section.

35. Conclusion section

Page 13, Lines 370-371: "broad indium-exposed population"

In this study, 57 subjects were included in the study, so I do not think it can be "broad". Please delete it or replace it with a different word.

36. Typo and unformal word

Page 6, Line 169: "p-val-ues": Is it typo of "p-values" ?

Page 13, Line 347: "correlationship": The word does not appear to be a formal expression (e.g., it is not listed in the Oxford dictionary). I recommend replacing it with "correlation".

7. PLOS authors have the option to publish the peer review history of their article (what does this mean?). If published, this will include your full peer review and any attached files.

Reviewer #1: No

Reviewer #2: **Yes: **Toshiharu Mitsuhashi

---

## [Author Response · Author response to Decision Letter 1]

19 Oct 2020

Responses to Academic Editor:

Thank you for the opportunity to revise my paper. I am honored to have the opportunity to talk about the indium levels measured by the subjects in the control group. Through consulting the previous literature, the author found that the geometric mean of indium in blood of general population was lower than the detection limit. Therefore, when the present results appeared, the author thought that there was something wrong with the detection. After careful analysis and verification, there is no problem with our detection method, and the selected control population has no previous occupational exposure. As a result, we analyzed that the possible reason was background indium exposure. The population in our research is in Liuzhou City, Guangxi Province, China, which is the richest and most abundant indium resources in China. They are mainly distributed along the river. Although there was no occupational indium exposure in the control population, the indium pollution in river or soil was not ruled out. If possible, we would like to study the serum or urine indium levels of local non occupational indium-exposed populations.

DOI：dx.doi.org/10.17504/protocols.io.bne2mbge

Response to Reviewers:

Response Reviewer #1: 

why the control subjects would have measurable indium in blood?

Thank you for the opportunity to talk about the indium levels measured by the subjects in the control subjects. Indium is not a trace element and the author found that the geometric mean of indium in blood of general population was lower than the detection limit by consulting the literature. But at the same time, there have not been large studies to measure blood indium in unexposed populations. Therefore, up to now, there are very few reports on the level of indium in normal human serum or blood. First of all, the author can confirm that the detection method in this paper is appropriate and the selected control population has no previous occupational exposure. As a result, we analyzed that the possible reason was background indium exposure. The population in our research is in Liuzhou City, Guangxi Province, China, which is the richest and most abundant indium resources in China. Guangxi has become the largest indium production base in China due to its advantaged indium resources, and is known as "indium capital" in the world. Those indium bearing mine are mainly distributed along the river and there are many local production and processing enterprises. Although there was no occupational indium exposure in the control population, the indium pollution in river or soil was not ruled out. Of course, this is only our speculation. If possible, we would like to study the serum or urine indium levels of local non-occupational indium-exposed populations.

Response Reviewer #2: 

Thank you for the opportunity to revise my paper. An explanation of the points is below:

1.*3-r1. multiplicity of statistical tests

Thank you very much for your detailed description of the statistical problems in our study. The inevitable limitation of our study is that the sample size is too small. In order to distinguish between cases where there is a truly significant difference and cases where the difference appears to be significant by type I errors, the author reduced the number of calculated p-values. Therefore, the author makes some modifications in the analysis of the Results section (Table 5).

2.*5-r1. Check List

Thank you for sending me a link to the STROBE checklist. The author has carefully read each checklist item of the STROBE checklist to use with this article.

3.*10-r1. standardized regression coefficient

The author has added and explained this description in the Method section.

4.*16-r1. personal protective equipment

The author has explained this in the Method section (Subject). 

5.*17-r1. Control subject

Information on the control subjects has been supplemented in the body of the manuscript.

The author has added an explanation about the selection criteria of the study population and added an explanation for the difference in the percentage of smokers between the groups.

6.*18-r1 Comparison between workers and control

The author has added this Table for the manuscript.

[New comments]

7.Limitation

The author has added the limitations of this study in the Abstract and Discussion section.

8.Duplicate expressions.

The author has merged the duplicate expressions.

9. emotional expression

The author has changed the way of expression.

10.Description to be included in the Result section

The author has put the description of results in the Results section

11.Conclusion section 

The author has deleted “broad”.

12.Typo and unformal word

I'm very sorry. This is the author's mistake. The author has corrected it.

---

## [Decision Letter · Decision Letter 2]

2 Nov 2020

PONE-D-20-20542R2

Biomonitorization of Concentrations of 28 Elements in Serum and Urine among Workers Exposed to Indium Compounds

PLOS ONE

Dear Dr. yao,

Thank you for submitting your manuscript to PLOS ONE. Please fully address the reviewer's concerns before submitting the revised manuscript for re-consideration. 

We look forward to receiving your revised manuscript.

Kind regards,

Yi Hu

Academic Editor

PLOS ONE

Reviewers' comments:

Reviewer's Responses to Questions

**Comments to the Author**

1. If the authors have adequately addressed your comments raised in a previous round of review and you feel that this manuscript is now acceptable for publication, you may indicate that here to bypass the “Comments to the Author” section, enter your conflict of interest statement in the “Confidential to Editor” section, and submit your "Accept" recommendation.

Reviewer #2: (No Response)

2. Is the manuscript technically sound, and do the data support the conclusions?

Reviewer #2: Yes

3. Has the statistical analysis been performed appropriately and rigorously? 

Reviewer #2: No

4. Have the authors made all data underlying the findings in their manuscript fully available?

Reviewer #2: Yes

5. Is the manuscript presented in an intelligible fashion and written in standard English?

Reviewer #2: Yes

6. Review Comments to the Author

Reviewer #2: Please check the attached PDF file in detail.

Thank you for the second revision of the manuscript. More revisions have been made in more places and we believe that the research will be more qualified to the reader.

However, we still have concerns about some of the comments. Also, there are some points that should be pointed out in the revised manuscript. We ask that you please review and revise the manuscript.

It would also help me to complete this review more quickly if the rebuttal letter could be more specific as to which parts (Pages and Lines) of the manuscript have been changed.

7. PLOS authors have the option to publish the peer review history of their article (what does this mean?). If published, this will include your full peer review and any attached files.

Reviewer #2: **Yes: **Toshiharu Mitsuhashi

---

## [Author Response · Author response to Decision Letter 2]

30 Dec 2020

Responses to Academic Editor:

Thank you for the opportunity to revise my paper. 

Response to Reviewers:

Response Reviewer #2: 

Thank you for the third opportunity to revise my paper. An explanation of the points is below:

[Ongoing concerns]

*3-r2. Multiple statistical testing Problem (False positive rate will be extremely high)

Thank you very much for your careful explanation and explanation of this problem, so that the author has the opportunity to learn a lot of analysis methods. 

The author carefully interpreted the p value, added that “this is an exploratory study” in the Method section, added it as a Limitation in the Discussion section. 

*13-r2. Are the study subjects exposed to any other toxic metals?

The selection criteria for the study population in this study were indeed no history of occupational exposure to other metals. The author’s response stated "the indium-exposed workers were exposed to other toxic metals, too." , which means that lead, arsenic, cadmium and other toxic metals exist in the process of indium smelting, workers may also be exposed to lead, arsenic, cadmium and other toxic metals when they are exposed to indium. However, this does not mean that the indium-exposed workers we chose are also occupationally exposed to other toxic metals.

*17-r2 How to select the study subjects 

Please see the flow diagram.

*17-r2-(1). How to select non-exposed persons

I'm sorry, maybe there is something wrong with my language expression in my rebuttal letter (R1). The study subjects (= Indium-exposed workers + unexposed workers) are not volunteers. The Indium-exposed workers were selected from an indium ingot production plant (Guangxi, China) who were mainly exposed to indium metal. The unexposed workers (Control subjects) were a random sample of all volunteers derived from another nearby factory, who were office workers and had no history of occupational exposure to indium and other metals. The selected control subjects were matched to the age, sex distribution, average employment history, smoking status and drinking habits with indium-exposed workers.

Respond to the clarification noted in Rev-com(R1) comment No *17-r1: The study subjects were initially matched by inclusion criteria, the number of person is 69:69; but after a series of exclusion criteria, there were some differences between indium-exposed workers and control subjects (the number of person is 57:63).

*17-r2-(2). Number of excluded subjects.

The author described the number of subjects excluded because of criteria (3) and (4).

*17-r2-(3). Definition of Outlier

In our analysis of serum elements of all study subjects, we found that, the serum magnesium level of one control worker was abnormally low (3.8 mg/L). But the worker did not know about his problem. After our discovery, he went to the hospital for further examination and found that he had hypomagnesemia. Thus observations in this worker who had outliers outcome were excluded.

*17-r2-(4). Need for flow diagram

The author saw the explanation in STROBE 13(c) Consider the Use of a Flow Diagram; and supplemented a flow diagram.

Figure 1. Flow of Indium-exposed workers and unexposed workers in the study.

*17-r2-(5). Selection Bias and Generalizability

The author added “Potential confounders and sources of bias” in the Method section, added “Limitations and Generalization of the study results” in the Discussion section.

[Concerns of the newly additional parts]

37. about Table 2.

What the author wants to express is "employment period (years)". Changes have been made in Table 2.

38. about Table 5.

After checking and verifying Table 3, the author found some errors and corrected them in Table 5.

39. position of the Limitation

The author has listed the limitations in the Discussion section.

40. the Terms used in the selection criteria.

The author has changed the word "parameter" to "measured value".

---

## [Decision Letter · Decision Letter 3]

18 Jan 2021

PONE-D-20-20542R3

Biomonitorization of Concentrations of 28 Elements in Serum and Urine among Workers Exposed to Indium Compounds

PLOS ONE

Dear Dr. yao,

Thank you for submitting your manuscript to PLOS ONE. After careful consideration, we feel that it has merit but does not fully meet PLOS ONE’s publication criteria as it currently stands. Therefore, we invite you to submit a revised version of the manuscript that addresses the points raised during the review process.

We look forward to receiving your revised manuscript.

Kind regards,

Yi Hu

Academic Editor

PLOS ONE

Reviewers' comments:

Reviewer's Responses to Questions

**Comments to the Author**

1. If the authors have adequately addressed your comments raised in a previous round of review and you feel that this manuscript is now acceptable for publication, you may indicate that here to bypass the “Comments to the Author” section, enter your conflict of interest statement in the “Confidential to Editor” section, and submit your "Accept" recommendation.

Reviewer #2: (No Response)

2. Is the manuscript technically sound, and do the data support the conclusions?

Reviewer #2: Yes

3. Has the statistical analysis been performed appropriately and rigorously? 

Reviewer #2: Yes

4. Have the authors made all data underlying the findings in their manuscript fully available?

Reviewer #2: Yes

5. Is the manuscript presented in an intelligible fashion and written in standard English?

Reviewer #2: Yes

6. Review Comments to the Author

Reviewer #2: Thank you for the third revision of the manuscript. More revisions have been made in more places and we believe that the research will be more qualified to the reader.

The major concerns have already been fixed, but I have only two comments on minor points.

[Minor concerns]

41. The English word "volunteer" should be avoided.

According to the Rebuttal letter, it states that the research subjects are not volunteers. However, in the text, there is the following statement.

(Page 4, Lines 10-11, Study population Section)

"The control subjects in this study were a random sample of all volunteers derived from another nearby factory ,"

The term "volunteers" is used here. If they are not volunteers, then another word should be used.

Self-selection bias may occur in studies using volunteers and should be discussed in the text. To make it clear that this study does not require that consideration, I would recommend the English word "volunteer" should be avoided.

For example, "candidates" would be one choice of word to replace it.

42. Elemental symbols or English words.

With the exception of indium, most elements are described in the main text using element symbols. For example, "Al" is used instead of "aluminum".

However, in the following sections, English words are used instead of element symbols.

(Page 15, Lines 11-13, Result section)

"Selenium declined in serum (19%) ... But declined in urine (31%, 19%)."

To avoid confusion for the reader, I would recommend using the elemental symbols instead of the English words as in the other sections.

7. PLOS authors have the option to publish the peer review history of their article (what does this mean?). If published, this will include your full peer review and any attached files.

Reviewer #2: **Yes: **Toshiharu Mitsuhashi

---

## [Author Response · Author response to Decision Letter 3]

27 Jan 2021

Reply to revision

Responses to Academic Editor:

Thank you for the opportunity to revise my paper. 

Response to Reviewers:

Response Reviewer #2: 

Thank you very much for your patient guidance and careful revision of this paper, so that the author has the opportunity to learn a lot of writing and analysis methods. An explanation of the points is below:

[Minor concerns]

41. The English word "volunteer" should be avoided.

The author has changed the word "volunteer" to "candidates".

42. Elemental symbols or English words.

The author has used the elemental symbols instead of the English words.

---

## [Editor Report · Decision Letter 4]

29 Jan 2021

Biomonitorization of Concentrations of 28 Elements in Serum and Urine among Workers Exposed to Indium Compounds

PONE-D-20-20542R4

Dear Dr. yao,

We’re pleased to inform you that your manuscript has been judged scientifically suitable for publication and will be formally accepted for publication once it meets all outstanding technical requirements.

Kind regards,

Yi Hu

Academic Editor

PLOS ONE
---

## [Editor Report · Acceptance letter]

11 Feb 2021

PONE-D-20-20542R4 

Biomonitorization of Concentrations of 28 Elements in Serum and Urine among Workers Exposed to Indium Compounds 

Dear Dr. Yao:

I'm pleased to inform you that your manuscript has been deemed suitable for publication in PLOS ONE. Congratulations! Your manuscript is now with our production department. 

Kind regards, 

on behalf of

Prof. Yi Hu 

Academic Editor

PLOS ONE